# Latent Spatial Dirichlet Allocation

**Junsouk Choi**
Department of Biostatistics
University of Michigan
Ann Arbor, MI 48109
junsouk@umich.edu

**Jian Kang**
Department of Biostatistics
University of Michigan
Ann Arbor, MI 48109
jiankang@umich.edu

**Veerabhadran Baladandayuthapani**
Department of Biostatistics
University of Michigan
Ann Arbor, MI 48109
veerab@umich.edu

## Abstract

We propose a novel topic modeling approach, latent spatial Dirichlet allocation (LSDA), which generalizes the latent Dirichlet allocation to spatial data. LSDA integrates spatial Gaussian processes within the LDA framework, thereby effectively capturing complex spatial dependencies inherent in spatial data. We develop an efficient Markov chain Monte Carlo algorithm, and applications to both real and synthetic datasets successfully demonstrate the utility of LSDA.

## 1 Introduction

Latent Dirichlet allocation (LDA, Blei *et al.*, 2003) is a Bayesian probabilistic model commonly used in natural language processing to discover abstract topics that occur in a corpora of documents. The LDA model assumes that the words of each document arise from a mixture of topics, where each topic is a multinomial distribution over a fixed word vocabulary, suggesting a shared theme in the data. Documents consist of multiple topics with varying proportions, implying that LDA is an instance of mixed-membership models. LDA is a powerful tool in modeling hidden thematic structures in a collection of documents, facilitating the organization and understanding of large volumes of text data. The mixed-membership nature of the LDA model makes it popular in analyzing discrete data in other settings, such as population genetics (Pritchard *et al.*, 2000), network data (Airoldi *et al.*, 2008) and computer vision (Li & Perona, 2005; Russell *et al.*, 2006).

However, the direct application of LDA or conventional LDA-based topic models to spatial data analysis faces several challenges. First, LDA is a 'bag of words' model, which means that the words in each document are assumed to be exchangeable within them. This assumption causes LDA to ignore spatial structures within documents. Second, LDA assumes that topic distributions are independent across documents. In the context of spatial analysis, where documents typically represent segments of the entire spatial region, this assumption, along with the bag-of-words assumption, completely disregards the spatial dependence structure inherent in spatial data. In practice, we expect that topic distributions in neighboring spatial regions are correlated. To address such limitations, several extensions of the LDA model have been developed to account for spatial information in diverse fields of spatial data analysis. For instance, in the context of location-based social network data, spatial topic models have been developed to identify regional communities (Van Canh & Gertz, 2013) and assign semantic labels to locations of interest (He *et al.*, 2017). In computer vision, topic models such as the spatial LDA model (Wang & Grimson, 2007) and the spatially coherent latent topic model

Workshop on Bayesian Decision-making and Uncertainty, 38th Conference on Neural Information Processing Systems (NeurIPS 2024).

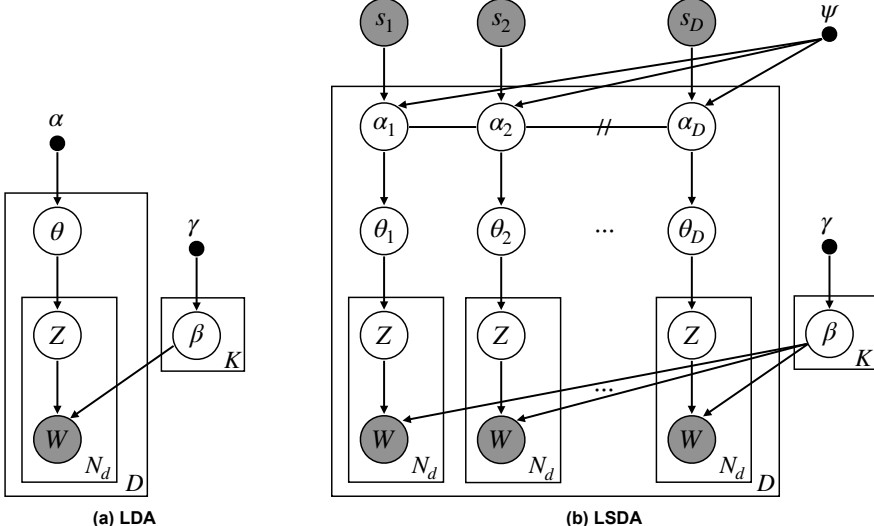

Figure 1: Probabilistic graphical representations of the conventional LDA (a) and the proposed LSDA (b).

(Cao & Fei-Fei, 2007) have been proposed to incorporate spatial information, allowing the discovery of object and scene categories from images. Chen *et al.* (2020) extended LDA to account for spatial structure in multiplex imaging data from human tissues. Despite their advancements, these methods are specifically designed for particular applications, which limits their generalizability across broad spatial data analytic tasks.

We propose the latent spatial Dirichlet allocation (LSDA), a general topic modeling framework that can be applied or easily extended to a broad range of spatial data classes and dependencies. Spatial data (e.g., a multiplex image in Figure 2(a)) serve as input for the proposed LSDA method, which in turn generates outputs that include spatial patterns of topics across the spatial region and estimates of topics themselves (e.g., Figure 2(b)). Specifically, the proposed LSDA employs spatial Gaussian processes (GPs) to incorporate spatial information into topic modeling, providing great flexibility to model the inherent spatial correlation structures within spatial data. We also develop an efficient Markov chain Monte Carlo (MCMC) algorithm for performing posterior inference for the LSDA model, and demonstrate its effectiveness through applications to both real cancer imaging as well as synthetic datasets.

## 2   LSDA Topic Model

To generalize topic models to spatial data analysis, it is essential to appropriately define "words" and "documents" in the context of spatial data. Any low-level spatial objects characterized by discrete values and observable locations can be considered as words. For instance, in the multiplex imaging data shown in Figure 2(a), individual cells are observed with their specific types and locations. These cells can be considered as the words for our LSDA model. For the definition of documents, we consider distinct segments of the overall spatial region, where the locations of the segments are observable. The segmentation of the spatial domain and the determination of the number of segments are pre-processing steps that vary depending on the specific application. For example, in multiplex imaging data analysis, biological prior knowledge can guide the segmentation process. In the application presented in Section 3.1, we defined spatial segments based on the biological insight that cell-to-cell interactions are typically negligible beyond a certain distance. In computer vision tasks, spatial segments could be designed to include sufficient local pixels to capture relevant local features while remaining smaller than object sizes in images. Let $D$ denote the number of the spatial segments, and $w_{di} \in \{1, \ldots, W\}, i = 1, \ldots, N_d$, represent the multiple words within each segment, where $N_d$ is the number of words in the $d$-th spatial segment of the entire region.

**Latent Dirichlet Allocation.** LDA is a Bayesian hierarchical model for a collection of documents (originally a corpora and, in our application, spatial data). Assume that there exist $K$ topics and

let $z_{di} \in \{1, \ldots, K\}$ be a topic assignment latently associated with each word $w_{di}$. LDA assumes the following generative process for a collection of documents. First, for each topic $k$, draw a distribution over the vocabulary from a Dirichlet distribution, $\beta_k \sim \text{Dirichlet}(\gamma)$. Next, for each document $d$, draw a distribution over topics from a Dirichlet distribution, $\theta_d \sim \text{Dirichlet}(\alpha)$. Then, for each word $i$ in document $d$, draw a topic index from the document-specific topic distribution, $z_{di} \sim \text{Multinomial}(\theta_d)$, and draw the observed word from the word distribution corresponding to the selected topic $z_{di}$, $w_{di} \sim \text{Multinomial}(\beta_{z_{di}})$. Here, $\gamma$ and $\alpha$ are hyperparameters for the Dirichlet distributions. A graphical representation of the generative process of LDA is shown in Figure 1(a). A fundamental assumption of the LDA model is that the documents are independent of each other. However, this assumption does not hold for spatial data, as topic distributions of nearby spatial regions are expected to be correlated.

**Latent Spatial Dirichlet Allocation.** We generalize the LDA model to incorporate the spatial information inherent in the spatial data. Specifically, to account for spatial dependence between different segments of the entire spatial region, we introduce the spatial Dirichlet allocation process (SDAP) for the segment-specific topic distribution $\theta_d := \theta(s_d) = \{\theta_1(s_d), \ldots, \theta_K(s_d)\}^\top \in \mathbb{R}^K$, where $s_d$, $d = 1, \ldots, D$, denote the observed locations of spatial segments. We define the SDAP on $\theta(\cdot)$ as

$$\theta(\cdot)|\alpha(\cdot) \sim \text{Dirichlet}\{\alpha(\cdot)\}, \qquad \log \alpha_k(\,\cdot\,) \overset{iid}{\sim} \mathcal{GP}\{0, \kappa(\cdot, \cdot)\}, \ k = 1, \ldots, K,$$

where $\alpha(\cdot) = \{\alpha_1(\cdot), \ldots, \alpha_K(\cdot)\}^\top$ and $\kappa$ is the covariance kernel for the GPs. Here, at the location $s_d$ of the $d$-th segment, $\alpha(s_d) \in \mathbb{R}^K$ serves as the hyperparameters for the Dirichlet prior on the segment-specific topic distribution $\theta_d = \theta(s_d)$. Since $\alpha(\cdot)$ is spatially smoothed by the use of GPs, if two spatial segments are nearby with locations $s_{d_1}$ and $s_{d_2}$, the corresponding Dirichlet hyperparameters $\alpha(s_{d_1})$ and $\alpha(s_{d_2})$ are likely to be similar. Since they determine the mean of Dirichlet distribution with $E\{\theta_k(s_d)|\alpha(s_d)\} = \alpha_k(s_d)/\sum_{k=1}^K \alpha_k(s_d)$, the similarity in $\alpha(s_{d_1})$ and $\alpha(s_{d_2})$ will result in similar segment-specific topic distributions $\theta(s_{d_1})$ and $\theta(s_{d_2})$. Hence, the distribution over topics will be coherent across neighboring spatial regions. An illustrative example of the proposed SDAP is provided in Appendix A.

The SDAP represents a novel spatial stochastic process for modeling spatially varying topic distributions within the topic modeling paradigm. It contrasts with related models such as the spatial Dirichlet process mixture model (SDPM, Gelfand *et al.*, 2005), which employs a Dirichlet process mixture with a GP baseline to generate a random spatial process. Unlike the SDPM, which employs the Dirichlet process to model continuous spatial data, our SDAP relies on a finite Dirichlet distribution, producing spatially varying discrete distributions over topics rather than continuous spatial outcomes.

We denote our spatial Dirichlet allocation process as $\theta(\cdot) \sim \text{SDAP}(\kappa)$. Then, the complete data generating process of LSDA is given as follows. The first step is same with the standard LDA. For $k = 1, \ldots, K$, draw per-topic word distributions from a Dirichlet distribution, $\beta_k \sim \text{Dirichlet}(\gamma)$. However, the procedure diverges in the subsequent step. For the segments $d = 1, \ldots, D$ of the entire spatial region, we set their topic distributions at $\theta_d = \theta(s_d)$, where $\theta(\cdot) \sim \text{SDAP}(\kappa)$. The spatial Dirichlet allocation induces spatially coherent topic distributions across neighboring spatial locations. Given $\beta_k$'s and $\theta_d$'s, for word $i$ in segment $d$, we draw a topic index from $z_{di} \sim \text{Multinomial}(\theta_d)$, and draw the observed word from the selected topic $z_{di}$, $w_{di} \sim \text{Multinomial}(\beta_{z_{di}})$. A probabilistic graphical representation of the proposed LSDA is provided in Figure 1(b).

**Posterior Computation of LSDA.** For posterior inference on the proposed LSDA, we develop a computationally efficient MCMC algorithm to simulate the joint posterior distribution of all model parameters, including $\log \alpha_k(\cdot)$, to which we assign a GP prior. A major challenge in our posterior computation is that sampling $\log \alpha_k(\cdot)$ from the GP posterior requires substantial computing resources because it requires us to invert increasingly large covariance matrices as the number of documents (i.e. spatial locations) increases. To address this problem, we adopt an approximation of the GP through the eigendecomposition of the covariance kernel. According to the Karhunen–Loève theorem, $\log \alpha_k(s)$, which follows a GP, can be equivalently represented as a linear combination of the eigenfunctions. Given the spatial smoothness assumption of $\log \alpha_k(\cdot)$ in our LSDA framework, we require only a limited number of eigenfunctions $L$, much smaller than the number of spatial segments $D$, to adequately approximate $\log \alpha_k(\cdot)$. This reduction leads to efficient posterior computation. Therefore, leveraging this GP approximation, we develop a Metropolis-Hastings within Gibbs sampler for our GP-based LSDA model. Our sampler utilizes Gibbs sampling for updating $\beta_k$ and $\theta_d$, while collapsed Gibbs sampling is employed for updating $z_{ij}$. The basis coefficients for the eigenfunctions, which

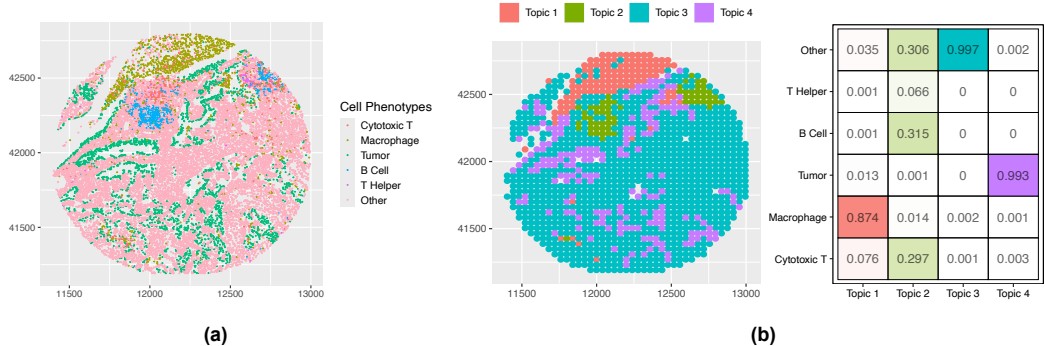

Figure 2: (a) Selected ovarian cancer image. (b) LSDA results: Dominant topics across spatial segments of the ovarian cancer image, along with estimated topics $\hat{\beta}_k$.

approximate the GPs, are updated using the stochastic gradient Hamiltonian Monte Carlo approach proposed by Chen *et al.* (2014). Further details of our MCMC sampling schemes are provided in Appendix B.

## 3    Experiments

### 3.1    Application to Cancer Multiplex Imaging Data

We demonstrate the utility of the proposed LSDA with a multiplex imaging dataset of high grade serous ovarian cancer (Steinhart *et al.*, 2021). The ovarian cancer multiplex imaging dataset consists of segmented and phenotyped multiplex images of tumor regions collected from 128 patients with ovarian cancer. In each image, the spatial locations and types of tumor and immune cells were observed, with cells in the dataset classified into $W = 6$ types: Cytotoxic T cell, Macrophage, Tumor cell, B cell, T Helper cell, and Other. We aimed to uncover spatial tissue architecture by identifying biologically significant tumor microenvironments where specific cell types co-localize.

We applied our proposed LSDA to an image with the highest cell count that also contained at least 30 cells per type, shown in Figure 2(a). Since the number of topics was unknown in this analysis, we considered multiple numbers of topics $K = 2, 3, 4, 5$, and utilized the deviance information criterion (DIC, (Li *et al.*, 2020)) to select the optimal number of topics. According to the DIC, $K = 4$ was selected. Figure 2(b) displays the estimates of topics $\hat{\beta}_k, k = 1, \ldots, 4$, and the dominant topics across spatial segments, identified by the highest topic probability estimate $\hat{\theta}_{kd}$ across segments. Detailed information on the implementation of our LSDA approach can be found in Appendix C. In this application, topics represents the co-localization of various cell types with varying degrees, each corresponding to a distinct tumor microenvironment. Specifically, Topic 2 indicates a microenvironment where B cells and cytotoxic T cells co-localize. In ovarian cancer, this aggregation is known to enhance tumor immunity and improve patient survival, as B cells act as antigen-presenting cells that help activate cytotoxic T cells and contribute to the antitumor response (Montfort *et al.*, 2017; Zhang *et al.*, 2023).

To evaluate our LSDA, we compared our results with two benchmarks: standard LDA (Blei *et al.*, 2003), which does not incorporate spatial information, and Spatial-LDA (Chen *et al.*, 2020), which leverages an adjacency-based spatial similarity prior to account for spatial structures in multiplex imaging data. The results of applying LDA and Spatial-LDA with $K = 4$ to the same ovarian cancer image are shown in Figures 5 and 6 (Appendix C). Our LSDA demonstrated clear advantages in analyzing cancer multiplex imaging data. Compared to the benchmarks, LSDA better captured diverse co-localizations of cell types while maintaining spatial coherence of the estimated topics across tissue regions. Notably, only LSDA successfully identified the co-localization of B cells and cytotoxic T cells (Topic 2 in Figure 2(b)), a tumor microenvironment that neither LDA nor Spatial-LDA was able to capture. This highlights the superior flexibility of LSDA in uncovering spatially correlated hidden tumor microenvironments from multiplex imaging data.

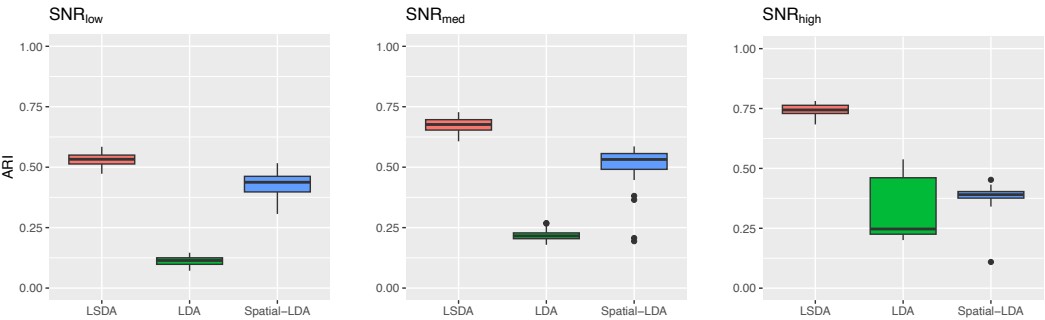

Figure 3: Boxplots of ARIs for true vs. estimated dominant topics across 50 synthetic data replicates for LSDA, LDA, and Spatial-LDA under different SNR scenarios.

## 3.2 Synthetic data

We also conducted a comparative evaluation of the empirical performance of LSDA using synthetic data generated based on the real data discussed in Section 3.1. Synthetic datasets were generated from the LSDA model using fixed cell locations from the ovarian cancer multiplex image, with $W = 6$ and $K = 4$ as in Section 3.1. We set $\alpha_k = \hat{\alpha}_k$ (fitted GPs from the ovarain data), and considered three different settings of $\beta_k$ corresponding to varying signal-to-noise ratios (SNRs): $\beta_k = (1-\rho)\hat{\beta}_k + \rho(\frac{1}{W}\mathbf{1}_W)$, where $\hat{\beta}_k$ is the estimate from the ovarian data and $\mathbf{1}_W$ is a vector of ones of length $W$. As $\rho$ increases, topics get less distinct, indicating the SNR decreases. We varied $\rho$ across three settings: 0.5, 0.25, and 0, labeled as $SNR_{low}$, $SNR_{med}$, and $SNR_{high}$, respectively.

We implemented our LSDA, and compared its performance against the standard LDA and Spatial-LDA as in Section 3.1. The effectiveness of these methods in identifying spatial patterns of topics was compared by calculating the adjusted rand index (ARI, Hubert & Arabie, 1985) between the true and estimated segment-specific dominant topics for each method. Detailed information about our synthetic data experiment, including the calculation of the ARI, is provided in Appendix C. In Figure 3, we present boxplots of the ARI values (from 50 replicates) for LSDA, LDA, and Spatial-LDA across different SNR levels. LSDA consistently outperformed all the alternatives in identifying dominant topics. Additionally, as the SNR decreased, the performance gap between models that account for spatial dependencies (LSDA and Spatial-LDA) and the standard LDA model widened, which emphasizes the importance of accounting for spatial dependence structures when applying topic models to spatial analysis.

## 4 Conclusion

We propose LSDA, a novel topic modeling framework for spatial data, which integrates GPs within the LDA model to effectively account for complex spatial dependencies inherent in such data. Through simulations, we demonstrate that incorporating GPs allows for greater flexibility in modeling complex spatial dependency patterns, outperforming existing spatial topic models that rely on a specific assumption of spatial smoothness (e.g., adjacency-based similarity). Our analysis of cancer multiplex imaging data shows the effectiveness of our LSDA in discovering tumor microenvironments with biological significance. Given its flexibility, the proposed LSDA framework can also be applied to a wide range of spatial data, including computer vision and geographical data.

Currently, LSDA focuses on categorical spatial data, where each spatial unit is assigned a categorical label. The current framework is not directly applicable to continuous spatial data; while discretization of continuous data is a possible workaround, it inevitably leads to a loss of information. To address this, we plan to extend our model to accommodate continuous spatial data in future work. Another promising direction for future research is improving the scalability of the LSDA method. Although the current fully Bayesian approach, based on MCMC, naturally provides uncertainty quantification, it can be computationally intensive. To improve scalability, we plan to explore alternative posterior inference methods, such as variational Bayes.

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

## A  Illustration of the SDAP

This section presents an illustrative example of our proposed SDAP. Figure 4(a) displays realizations of $\log \alpha_k(\cdot) \sim \mathcal{GP}\{0, \kappa(\cdot, \cdot)\}, k = 1, \ldots, K$ with $K = 3$. Here, for the covariance kernel $\kappa$, we employ the modified squared exponential kernel described in Appendix C.2, with hyperparameters set to $a = 0.01$ and $b = 1$. Given the realizations of $\log \alpha_k(\cdot)$, Figure 4(b) demonstrates the expected spatially varying topic distribution, $E(\theta_k(\cdot)|\alpha(\cdot)), k = 1, 2, 3$, which reveals spatial coherence across neighboring regions. Taken together, this example illustrates how GP-based spatial smoothing applied to the spatially varying Dirichlet hyperparameters $\log \alpha_k(\cdot)$ induces spatial coherence in the segment-specific topic distributions across different regions.

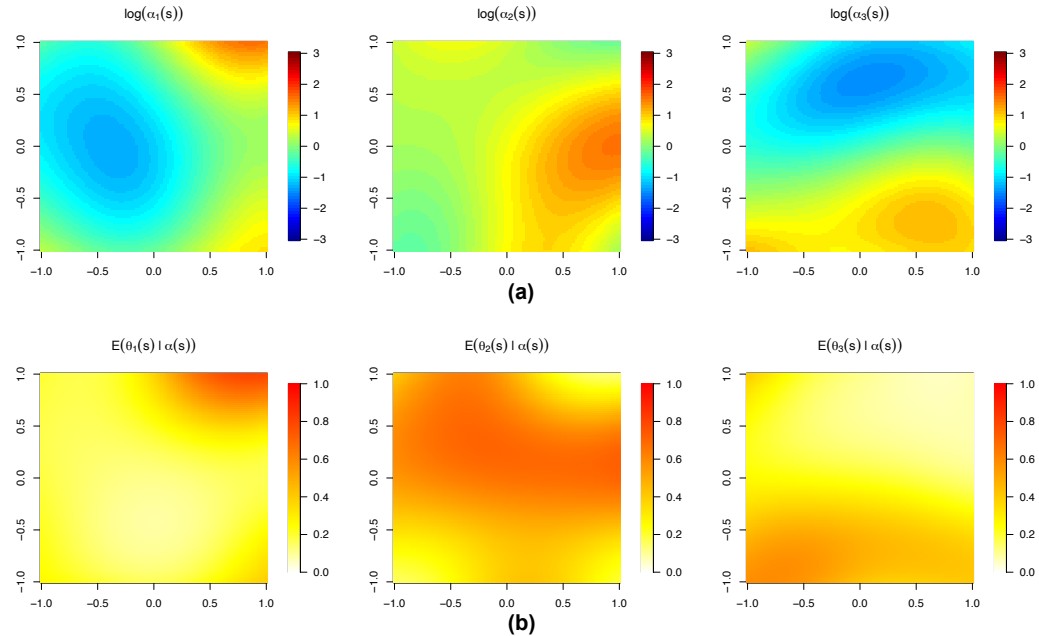

Figure 4: (a) Realizations of $\log \alpha_k(\cdot) \sim \mathcal{GP}\{0, \kappa(\cdot, \cdot)\}$ for $k = 1, 2, 3$. (b) Corresponding expected spatially varying distribution $E(\theta_k(\cdot)|\alpha(\cdot))$ for $k = 1, 2, 3$.

## B  The MCMC Algorithm

In this section, we provide the detailed description of our MCMC algorithm for the LSDA model. To alleviate the computational burden of sampling from the GP posterior, we consider an approximation of the GP using the eigendecomposition of the covariance kernel $\kappa$, as described below.

### B.1  GP Approximation

Consider the eigendecomposition of the covariance kernel $\kappa(s, s') = \sum_{l=1}^{\infty} \lambda_l \phi_l(s) \phi_l(s')$, where $\{\lambda_l\}_{l=1}^{\infty}$ is the set of eigenvalues with $\lambda_1 \geq \cdots \geq \lambda_l \geq \lambda_{l+1} \geq \cdots$, and $\{\phi_l(s)\}_{l=1}^{\infty}$ is the set of orthonormal eigenfunctions such that $\int \phi_l(s) \phi_{l'}(s) ds = 1(l = l')$ for any $l, l' \in \{1, 2, \ldots\}$. The Karhunen–Loève theorem implies that $\log \alpha_k(s)$, modeled as a GP, can be equivalently represented

as a linear combination of the eigenfunctions, $\log \alpha_k(s) = \sum_{l=1}^{\infty} b_{kl}\phi_l(s)$, where $b_{kl} \overset{iid}{\sim} \text{N}(0,1)$. Hence, we can approximate $\log \alpha_k(s)$ by truncating the summation at a sufficiently large number of components $L$: $\log \alpha_k(s) \approx \sum_{l=1}^{L} b_{kl}\phi_l(s)$. Since our LSDA model assumes $\log \alpha_k(\cdot)$ is spatially smooth, the required number of eigenfunctions $L$ to achieve a good approximation of $\log \alpha_k(\cdot)$ is still much smaller than the number of spatial segments $D$, leading to efficient posterior computation.

## B.2  Markov chain Monte Carlo

To fit the GP-based LSDA model, we develop a Metropolis-Hastings within Gibbs sampler to draw samples from the posterior distribution. For $\beta_k$ and $\theta_d$, the full conditional distributions have closed forms, leading to efficient Gibbs sampling update schemes. For $z_{ij}$, we use the collapsed Gibbs sampling, marginalizing out $\beta_k$'s and $\theta_d$'s from our target posterior distribution. Updating the basis coefficients $B = (b_{kl})_{k,l}$, which approximate GPs over $\log \alpha_k(\cdot), k = 1, \ldots, K$, is the most challenging part in our MCMC algorithm, because of their high-dimensionality and the complexity of the full conditional density that involves the log transformation. Therefore, we adopt the stochastic gradient Hamiltonian Monte Carlo (SGHMC) approach proposed by Chen *et al.* (2014) to update $B$. The detailed steps of our sampler are given as follows:

*Update Z.* Sample $z_{di}$ from its conditional posterior $Pr(z_{di} = \tilde{k}|Z_{-di}, B, X)$, marginalizing out $\beta$ and $\theta$,

$$Pr(z_{di} = \tilde{k}|Z_{-di}, B, X) \propto \frac{n_{\tilde{k}x_{di}}^{(-di)} + \gamma_{x_{di}}}{\sum_{w=1}^{W}(n_{\tilde{k}w}^{(-di)} + \gamma_w)} \frac{N_{d\tilde{k}}^{(-di)} + \theta_{\tilde{k}}(s_d)}{\sum_{k=1}^{K}(N_{dk}^{(-di)} + \theta_k(s_d))}, \quad (1)$$

where $n_{kw}$ is the number of times word $w$ is assigned to topic $k$, $N_{dk}$ is the number of words in document $d$ assigned to topic $k$, and the superscript $(-di)$ indicates that the corresponding datum has been disregarded when calculating $n_{kw}$ and $N_{dk}$.

*Update $\beta$.* Sample $\beta_k$ independently from its full conditional

$$\pi(\beta_k|Z, X) \propto \text{Dirichlet}\left(n_{k1} + \gamma_1, \ldots, n_{kW} + \gamma_W\right). \quad (2)$$

*Update $\theta$.* Sample $\theta_d$ independently from its full conditional

$$\pi(\theta_d|Z, B, X) \propto \text{Dirichlet}\left(N_{d1} + \alpha_1(s_d), \ldots, N_{dK} + \alpha_K(s_d)\right), \quad (3)$$

where $\alpha_k(s_d) = \exp\left(\sum_{l=1}^{L} b_{kl}\phi_l(s_d)\right)$.

*Update B.* It is infeasible to directly sample from the full conditional distribution of $B$

$$\pi(B|\theta, X) \propto \prod_{d=1}^{D} \frac{1}{\text{B}\left(\alpha_1(s_d), \ldots, \alpha_K(s_d)\right)} \theta_{d1}^{\alpha_1(s_d)-1} \cdots \theta_{dK}^{\alpha_K(s_d)-1} \prod_{k=1}^{K} \prod_{l=1}^{L} \exp\left(-\frac{1}{2}b_{kl}^2\right), \quad (4)$$

where $\alpha_k(s_d) = \exp\left(\sum_{l=1}^{L} b_{kl}\phi_l(s_d)\right)$ and $\text{B}(c_1, \ldots, c_K)$ is the beta function. Hence, we adopt SGHMC (Chen *et al.*, 2014) to draw samples from (4). Hamiltonian Monte Carlo (HMC) is an efficient sampling approach which shows a higher acceptance rate compared to the standard Metropolis-Hastings sampling. SGHMC extends HMC by using stochastic gradients to improve efficiency, allowing it to avoid evaluating the entire dataset. Additionally, SGHMC eliminates the need for the Metropolis-Hastings step after each proposal by introducing an additional friction term in the momentum update. Specifically, at $t$-th MCMC iteration, we update $B$ through Algorithm 1.

Given the sample of other parameters at the $t$-th iteration $\theta_d^{(t)}$, we compute the stochastic gradient of $U(B) = -\log \pi(B|\theta, X)$ by subsampling the indices of documents, that is, we calculate

$$\frac{\partial \tilde{U}(B)}{\partial b_{kl}} = \frac{D}{|\mathcal{D}|} \sum_{d \in \mathcal{D}} \left\{ \psi\left(\alpha_k(s_d)\right) - \psi\left(\sum_{k=1}^{K} \alpha_k(s_d)\right) - \log(\theta_{dk}^{(t)}) \right\} \alpha_k(s_d)\phi_l(s_d) + b_{kl}, \quad (5)$$

where $\psi(\cdot)$ is the digamma function and the index set $\mathcal{D}$ is a random subset of the indices of documents (spatial segments) $\{d\}_{d=1}^{D}$.

**Algorithm 1** SGHMC for updating $B$ at the $t$-th MCMC iteration

---

1: **Input:** the number of leapfrog steps $M$, the learning rate $\eta$ and the momentum tuning parameter
   $(1 - \alpha)$.
2: optionally, resample momentum $r$ from the matrix normal distribution $r \sim MN_{K \times L}(0, I, \eta I)$;
3: set $(B_0, r_0) = (B^{(t-1)}, r^{(t-1)})$;
4: **for** $h = 1, \dots, M$ **do**
5:     update $B_h = B_{h-1} + r_{h-1}$;
6:     sample $\epsilon_h \sim MN_{K \times L}(0, I, 2\alpha\eta I)$;
7:     sample $\mathcal{D} \subset \{d\}_{d=1}^{D}$;
8:     update $r_h = (1 - \alpha)r_{h-1} - \eta\nabla\tilde{U}(B_h; \mathcal{D}) + \epsilon_h$;
9: **end for**
10: set $(B^{(t)}, r^{(t)}) = (B_M, r_M)$;

---

## C   LSDA Implementation for the Experiments

In this section, we provide additional results and details for the experiments in Section 3, including the implementation our LSDA approach on cancer multiplex imaging data.

### C.1   Segmentation of Multiplex Image

The first step in implementing our LSDA on the multiplex imaging data was segmenting the given multiplex image to define documents for our LSDA framework. We achieved the segmentation by applying Voronoi tessellation over a systemically arranged grid of points. Each grid point serves as the location of each segments (i.e., $s_d$), and the arrangement of this grid was designed to maintain inter-cellular distances within each spatial segment under 40 $\mu$m, a threshold beyond which cell-to-cell interactions are typically considered insignificant (Mohammed *et al.*, 2024). This process resulted in 1,328 spatial segments for the selected ovarian cancer image.

### C.2   Hyperparameters

To implement the proposed LSDA method, we should specify the covariance kernel $\kappa$ for the GP over $\log \alpha_k(\cdot)$. In this work, we used the modified squared exponential covariance kernel, which allows for straightforward computation of eigenfunctions and eigenvalues through the use of Hermite polynomials. The modified squared exponential covariance kernel is defined as

$$\kappa(s, s') = \exp\{-a(||s||_2^2 + ||s'||_2^2) - b||s - s'||_2^2\}, \tag{6}$$

where $|| \cdot ||_2$ denotes the Euclidean norm, and $a > 0$ and $b > 0$ are hyperparameters. When $\log \alpha_k(\cdot)$ follows a GP with mean zero and the modified exponential covariance kernel, the hyperparameter $a$ controls the rate at which the variance $\mathrm{Var}\{\log \alpha_k(s)\}$ decays relative to $\mathrm{Var}\{\log \alpha_k(\mathbf{0})\}$. On the other hand, the hyperparameter $b$ determines the smoothness of the GP; smaller values of $b$ result in smoother GPs. In our LSDA implementation for the experiments in Section 3, we set $a = 0.01$ and $b = 1$ as the default values. To further improve performance of the proposed method, these hyperparameters can also be selected using a suitable model selection criterion, such as the DIC (Li *et al.*, 2020).

### C.3   Posterior Inference

In Section 3, which includes applications to both real cancer imaging data and synthetic data, we run our proposed MCMC algorithm for 20,000 iterations, of which the first 10,000 iterations were discarded as burn-in. Based on the MCMC samples, we calculated the posterior mean $\hat{\beta}_k$ and $\hat{\theta}_d$ of $\beta_k$ and $\theta_d$, and identified the dominant topic for each spatial segment by determining which topic had the highest probability, that is, $\hat{k}_d = \mathrm{argmax}_k \hat{\theta}_{kd}$.

### C.4   Benchmarking on Cancer Multiplex Imaging Data

For comparison, we applied two baseline methods– standard LDA (Blei *et al.*, 2003) and Spatial-LDA (Chen *et al.*, 2020)–to the ovarian cancer image in Section 3.1, using the same number of

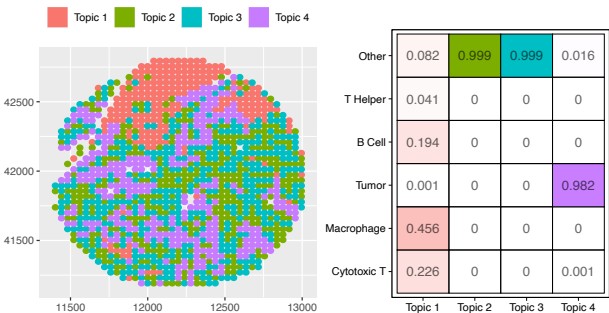

Figure 5: LDA results: Estimated topics and dominant topics with the highest posterior probabilities assigned to spatial segments of the ovarian cancer image.

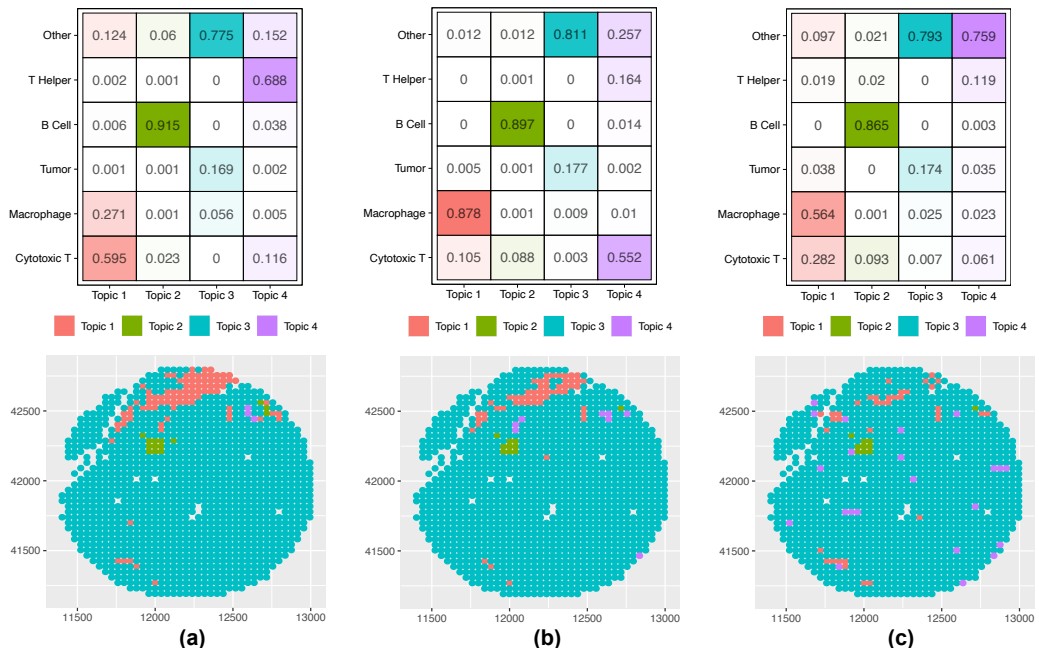

Figure 6: Spatial-LDA results: Estimated topics and dominant topic assignments across spatial segments for different values of the tuning parameter $d_{ij}$: $d_{ij} = 2.5 \times 10^{-2}$ (a), $d_{ij} = 2.5$ (b), and $d_{ij} = 2.5 \times 10^2$ (c).

topics $K = 4$ as in LSDA. Figure 5 illustrates the topics estimated by LDA and the dominant topic assignments across spatial segments, while Figure 6 presents the results obtained from Spatial-LDA. The Spatial-LDA approach relies on a tuning parameter $d_{ij}$ that controls the spatial similarity between adjacent regions, with smaller values of $d_{ij}$ implying stronger similarity in topic distributions. Thus, we evaluated Spatial-LDA with multiple values $d_{ij} \in \{2.5 \times 10^{-2}, 2.5, 2.5 \times 10^2\}$.

As expected, LDA produced a more scattered distribution of topics due to its its lack of spatial modeling capabilities. Although Spatial-LDA incorporates spatial information, the resulting clusters were predominantly dominated by a single topic enriched with "Other" cells, failing to yield informative spatial clustering patterns comparable to LSDA.

## C.5 Additional Details on the Synthetic Data Experiment

This section provides further details on our synthetic data experiment in Section 3.2. Specifically, for Spatial-LDA, we tested multiple values of the tuning parameter $d_{ij} \in \{2.5 \times 10^{-2}, 2.5, 2.5 \times 10^2\}$, consistent with Section 3.1, and present results for the setting that achieved the highest ARI.

In the synthetic data experiment in Section 3.2, the ARI (Hubert & Arabie, 1985) is defined by

$$\text{ARI} = \frac{\sum_{ij} \binom{D_{ij}}{2} - \left[\sum_i \binom{D_i}{2} \sum_j \binom{D_j}{2}\right] \bigg/ \binom{D}{2}}{\frac{1}{2}\left[\sum_i \binom{D_i}{2} + \sum_j \binom{D_j}{2}\right] - \left[\sum_i \binom{D_i}{2} \sum_j \binom{D_j}{2}\right] \bigg/ \binom{D}{2}},$$

where $n$ is the total number of spatial segments, $D_{ij}$ is the number of segments where the estimated dominant topic is $i$ and the true dominant topic is $j$, $D_i$ is the total number of segments assigned to the estimated dominant topic $i$, and $D_j$ is the total number of segments associated with the true dominant topic $j$. The ARI generally ranges from 0 to 1, where 1 indicates perfect agreement between the estimated and true clusterings (dominant topics), and 0 corresponds to random assignments.

