# OpenReview forum: "Latent Spatial Dirichlet Allocation"
_NeurIPS.cc/2024/Workshop/BDU — NeurIPS BDU Workshop 2024 Poster_

### Official Review · Reviewer_qaSL · 2024-09-22
**Good paper but can be made more solid**

**Rating:** 6
**Confidence:** 3

**Review:**

Summary: The authors propose Latent Spatial Dirichlet Allocation to capture the spatial structures present in the data by incorporating GPs to the traditional LDA model. The authors then develop efficient approximations for posterior inference of the proposed model. Finally, they apply the model to cancer multiplex imaging data as well as synthetic datasets to demonstrate the effectiveness of the model.

Strength: The paper is well-written, with a clear explanation of the methodology. The authors apply the proposed method to two cases, showcases its potential.

Weakness: 1. When applying the model to the cancer multiplex imaging data, the authors fail to compare with other baseline models such as LDA and Spatial-LDA. Including such results will highlight the advantages and gains of the proposed model. 2. The performance of the proposed model on synthetic data may not be entirely surprising, as the generative model for the synthetic data is identical to the proposed model itself. This raises questions about the fairness of the comparison, and additional evaluations on more varied synthetic datasets would strengthen the claims.

---

### Official Review · Reviewer_NMG5 · 2024-09-24
**I am leaning towards accepting the work for a workshop as it presents an interesting novel approach of using LDA for spatial data analysis. However, there are several open questions that should be addressed in the final version of the paper to improve clarity and completeness.**

**Rating:** 6
**Confidence:** 3

**Review:**

## Brief summary of the work
The authors propose the latent spatial Dirichlet allocation method, which is suitable for a broad range of spatial data classes and dependencies. The method is based on interpreting image data as being made out of different segments (=documents about topics) and each of those contains multiple words (=cells in the example). Each topics contains more of certain words.  Furthermore, the method is based on using a GP to sample the parameter alpha for the Dirichlet distribution for the segment-specific topic distribution. Thereby, they can account for spatial dependence between different segments. The whole model including GP is trained end-to-end as shown in Appendix A. The suitability of this approach is verified using a real and a synthetic dataset.

## Evaluation regarding quality
### Is the submission technically sound? Are claims well supported (e.g., by theoretical analysis or experimental results)?

The method is comprehensible using additionally the details provided in the appendix. However, additional information could be included for further clarification:
* The font size in Figure 1a is too small and difficult to read. Similarly, Figure 2 contains images that are too small. Increasing their size would enhance readability and make the results visible.
* Including illustrations of samples from the GP prior in the appendix would provide helpful visual context.
* The input to your GP consists of the spatial locations $s_d$​ of the $d$-th segment, meaning if $s_{d_1}$ and $s_{d_2}$​ are close to each other, the corresponding Dirichlet parameters alpha will also be close. However, how do you determine the number of segments?
* Is the GP trained or optimized during the LSDA model optimization as described in Appendix A? A brief sentence in the main paper indicating this would be beneficial. Initially, it seems as though the GP is an external model, and it wasn’t until much later that I realized it’s optimized alongside the LSDA model.
* In the appendix, you mention that default values are used for $a$ and $b$. Could these parameters be learned? This model is intriguing—learning $a$ and $b$ could enable its application across various datasets or domains. Does the GP tend to suffer from a reversion to the mean?

### Are the authors careful and honest about evaluating both the strengths and weaknesses of their work?
A short limitations section could be helpful.
### Are the methods used appropriate?
yes
### Is this a complete piece of work or work in progress?
A workshop paper

## Evaluation regarding clarity
### Is the submission clearly written?
Yes. Typo: line 10: arise
### Is it well organized? (If not, please make constructive suggestions for improving its clarity.)
Since Figures 2a and 2b are identical to Figure 1a, Figure 1a could be referenced rather than displaying the same figure twice. This would reduce redundancy and give you more space.
### Does it adequately inform the reader? (Note that a superbly written paper provides enough information for an expert reader to reproduce its results.)
Yes

## Evaluation regarding originality
### Are the tasks or methods new?
Yes, to the best of my knowledge.
### Is the work a novel combination of well-known techniques? (This can be valuable!)
Yes. LDA and GPs.
### Is it clear how this work differs from previous contributions?
As this is a workshop paper, there is no space to describe the methods of previous contributions. I would however suggest to do that in the main paper.
### Is related work adequately cited
Yes, except of: The citation in line 32: Chen et all (2014) seems incorrect. You might have intended to cite:
Chen, Zhenghao, et al. "Modeling multiplexed images with spatial-LDA reveals novel tissue microenvironments." Journal of Computational Biology 27.8 (2020): 1204-1218.

## Evaluation regarding significance
### Are the results important? Are others (researchers or practitioners) likely to use the ideas or build on them?
The method shows and new method of sampling alpha values for the spatial Dirichlet allocation process, introduced in the paper. Therefore, spatial GPs are used. The input to the GPs is spatial location information of pixels in image data. Pixels in image data are correlated, therefore, sampling the parameter alpha for the Dirichlet distribution for the spatial dependence between different segments using a GP will give similar alphas for similar pixels in the whole spatial region and different alpha values for distinct pixels.
Yes, this is in interesting approach to use LDA on spatial data. I would like to see the full paper version, after more experiments have been included and the method has been further refined.

### Does the submission address a difficult task in a better way than previous work? Does it advance the state of the art in a demonstrable way? Does it provide unique data, unique conclusions about existing data, or a unique theoretical or experimental approach?

How many segments did you use in your experiments, and how many grid points (i.e., spatial locations) did each segment have? Providing this information would help assess the computational efficiency of your GP model.
You mention that your method can be applied or easily extended to a wide range of spatial data classes and dependencies. In your experiments, you demonstrate its use on a multiplex imaging dataset and synthetic data derived from that. Have you tested it on other spatial datasets? Given the potential of this method, I believe there is room to evaluate the method on different datasets and get interesting results in future works, e.g. on datasets for segmentation.

## A list of its pros and cons
* Interesting idea of incorporating GPs to ensure nearby segment locations in documents/segments of the image are similar.
However, unclear how the number of documents gets defined.
* MCMC sampling well described in the appendix
* Interesting application: Multiplex imaging dataset of specific cancer cells and synthetic dataset considering various SNR values
However, it remains unclear how the method would generalize to other spatial data classes as claimed in line 37. Additionally, there is a lack of clarity on how to calculate ARI and how the baseline methods function. This might be understandable given that this is a workshop paper, but further elaboration would improve the paper's accessibility. Also, results not readable (Figures too small).
* Method well described with extensive appendix. Results show that method works well for the described approach.
However, no limitations section. Additionally, regarding line 70, for clarification: a spatial Dirichlet process is defined in the following reference:

Gelfand, Alan E., Athanasios Kottas, and Steven N. MacEachern. "Bayesian nonparametric spatial modeling with Dirichlet process mixing." Journal of the American Statistical Association 100.471 (2005): 1021-1035.

You mention that you introduce a spatial Dirichlet allocation process. How does your spatial Dirichlet allocation process differ from or compare to that one? Are you, in fact, introducing a new stochastic process or a method for sampling parameters for the Dirichlet distribution? Providing this clarification would help to understand the contribution, as you could then clearly point out what you did.

---

### Decision · Program_Chairs · 2024-10-09

Accept (Poster)